# Density of GABA_B_ Receptors Is Reduced in Granule Cells of the Hippocampus in a Mouse Model of Alzheimer’s Disease

**DOI:** 10.3390/ijms21072459

**Published:** 2020-04-02

**Authors:** Alejandro Martín-Belmonte, Carolina Aguado, Rocío Alfaro-Ruíz, Ana Esther Moreno-Martínez, Luis de la Ossa, José Martínez-Hernández, Alain Buisson, Ryuichi Shigemoto, Yugo Fukazawa, Rafael Luján

**Affiliations:** 1Synaptic Structure Laboratory, Instituto de Investigación en Discapacidades Neurológicas (IDINE), Dept. Ciencias Médicas, Facultad de Medicina, Universidad Castilla-La Mancha, Campus Biosanitario, C/Almansa 14, 02008 Albacete, Spain; Alejandro.Martin@uclm.es (A.M.-B.); Carolina.Aguado@uclm.es (C.A.); Rocio.Alfaro@uclm.es (R.A.-R.); AnaEsther.Moreno@uclm.es (A.E.M.-M.); drmartinezhernandez@gmail.com (J.M.-H.); 2Departamento de Sistemas Informáticos, Escuela Superior de Ingeniería Informática, Universidad de Castilla-La Mancha, 02071 Albacete, Spain; luis.delaossa@uclm.es; 3Grenoble Institut des Neurosciences, Université Grenoble Alpes, BP 170 Grenoble, France; alain.buisson@univ-grenoble-alpes.fr; 4Institute of Science and Technology (IST Austria), Am Campus 1, A-3400 Klosterneuburg, Austria; ryuichi.shigemoto@ist.ac.at; 5Division of Brain Structure and Function, Faculty of Medical Science, University of Fukui, Fukui 910-1193, Japan; yugo@u-fukui.ac.jp; 6Life Science Innovation Center, University of Fukui, Fukui 910-1193, Japan; 7Research Center for Child Mental Development, Faculty of Medical Science, University of Fukui, Fukui 910-1193, Japan

**Keywords:** Alzheimer´s disease, hippocampus, GABA_B_ receptors, ion channels, immunohistochemistry, electron microscopy, freeze-fracture, mouse model, dentate gyrus

## Abstract

Metabotropic γ-aminobutyric acid (GABA_B_) receptors contribute to the control of network activity and information processing in hippocampal circuits by regulating neuronal excitability and synaptic transmission. The dysfunction in the dentate gyrus (DG) has been implicated in Alzheimer´s disease (AD). Given the involvement of GABA_B_ receptors in AD, to determine their subcellular localisation and possible alteration in granule cells of the DG in a mouse model of AD at 12 months of age, we used high-resolution immunoelectron microscopic analysis. Immunohistochemistry at the light microscopic level showed that the regional and cellular expression pattern of GABA_B1_ was similar in an AD model mouse expressing mutated human amyloid precursor protein and presenilin1 (APP/PS1) and in age-matched wild type mice. High-resolution immunoelectron microscopy revealed a distance-dependent gradient of immunolabelling for GABA_B_ receptors, increasing from proximal to distal dendrites in both wild type and APP/PS1 mice. However, the overall density of GABA_B_ receptors at the neuronal surface of these postsynaptic compartments of granule cells was significantly reduced in APP/PS1 mice. Parallel to this reduction in surface receptors, we found a significant increase in GABA_B1_ at cytoplasmic sites. GABA_B_ receptors were also detected at presynaptic sites in the molecular layer of the DG. We also found a decrease in plasma membrane GABA_B_ receptors in axon terminals contacting dendritic spines of granule cells, which was more pronounced in the outer than in the inner molecular layer. Altogether, our data showing post- and presynaptic reduction in surface GABA_B_ receptors in the DG suggest the alteration of the GABA_B_-mediated modulation of excitability and synaptic transmission in granule cells, which may contribute to the cognitive dysfunctions in the APP/PS1 model of AD.

## 1. Introduction

The dentate gyrus (DG) is an integral part of the hippocampal formation and play key roles in the formation of episodic memory, spatial memory and exploration of new environments [1]. To fulfil these functions, the DG is highly organised and numerous synaptic connections take place between granule cells and basket cells, the principal cells and the primary inhibitory interneuron of the DG, respectively [1,2]. The major input of the DG arises from entorhinal cortex layer II neurons through the perforant path, and the functional integrity of this connection is necessary to fulfil its function in memory formation and spatial navigation [3]. Therefore, dysfunction in this entorhinal cortex–dentate gyrus pathway has been implicated in pathological conditions like Alzheimer´s disease (AD), the most prevalent neurodegenerative disease in the elderly population.

During progression of AD, neurodegeneration begins in the entorhinal cortex and then propagates to the adjacent regions, starting in the DG [4]. The three major neuropathology hallmarks of AD are extracellular fibrillary amyloid beta peptide (A*β*) in amyloid plaques, intraneuronal neurofibrillary tangles consisting of aggregated hyperphosphorylated tau and synapse loss [5]. Only loss of synapses has been correlated with cognitive decline and proposed to underlie learning and memory deficits in AD [6]. The vast majority of synapses in the DG of the hippocampus are glutamatergic excitatory synapses, and the main targets of these synapses are the dendritic spines of granule cells. Given the critical role of dendritic spines in memory, learning, and cognition, the loss of dendritic spines has been reported in the DG in AD [7,8,9] and in the APP/PS1 mouse model [10]. In addition, an impairment of adult neurogenesis has been reported in the DG in AD patients [11].

Granule cells in the DG are inhibited by γ-aminobutyric acid (GABA) synaptically released by interneurons. The response to GABA is mediated by activation of both ionotropic (GABA_A_) and metabotropic (GABA_B_) receptors [12,13]. The activation of GABA_B_ receptors is coupled to intracellular signal transduction mechanisms via G-proteins, mediating slow and prolonged synaptic inhibition through the activation of postsynaptic inwardly rectifying K^+^ channels and inhibition of presynaptic voltage-gated Ca^2+^ channels [13,14]. Acting on these effector ion channels, GABA_B_ receptors have modulatory actions on neuronal excitability and neurotransmitter release, and are involved in a number of physiological and pathophysiological processes [13]. Growing evidence supports the involvement of GABA_B_ receptors in AD [15,16,17,18,19].

Two different cDNAs encoding GABA_B_ receptor subunits have been cloned to date: GABA_B1_ and GABA_B2_ [20,21]. Both receptor subunits are required to form functional receptors, with the GABA_B1_ subunit being necessary for agonist activation and the GABA_B2_ subunit for surface trafficking and G-protein coupling [22,23,24,25]. Previous studies using different technical approaches showed that all sub-regions of the hippocampus, including the DG, express a high density of GABA_B_ receptors [20,26,27,28]. Immunoelectron microscopic studies demonstrated the main post- and pre-synaptic association of GABA_B_ receptors, mainly with excitatory synapses in dendritic spines of CA1 pyramidal cells and granule cells of the DG [28,29,30,31]. In addition, we have recently reported a dramatic reduction in GABA_B_ receptors along the postsynaptic plasma membrane of CA1 pyramidal cells, as well as at presynaptic sites in the excitatory synapses between Schaffer collateral-CA1 pyramidal cell spines in APP/PS1 mice [17]. Whether this alteration takes place in other principal cells of the hippocampus, such as granule cells, in the AD model mice is not yet known.

Given the loss of dendritic spines and excitatory synapses in the DG in APP/PS1 mice [10], it is expected that GABA_B_ receptors are also affected. Therefore, to identify possible alterations in the cellular and subcellular localisation of GABA_B_ receptors in this mouse model, we used high-resolution immunohistochemical techniques in combination with quantitative approaches at 12 months of age, a time when these animals show a large density of senile plaques and cognitive impairments [32,33] and when GABA_B_ receptors are altered along the surface of CA1 pyramidal cells [17]. Our data demonstrate for the first time a significant reduction in the surface expression of GABA_B_ receptors in the granule cells of the DG in APP/PS1 mice.

## 2. Results

### 2.1. Regional and Cellular Distribution of GABA_B1_ in the DG of Control and APP/PS1 Mice

We first investigated the regional and cellular distribution of GABA_B1_ in the DG in wild type and APP/PS1 mice at 12 months of age, using light microscopy immunohistochemical techniques (Figure 1). In both genotypes, immunoreactivity for GABA_B1_ was the highest in the molecular layer, weak in the hilus and weakest in the granule cell layer of the DG (Figure 1A–H). At the cellular level, GABA_B1_ immunoreactivity was weekly detected in the somata of granule cells but strongly expressed on their dendrites in the molecular layer at their default dendritic localisation (Figure 1C,D,F,G). GABA_B1_ immunoreactivity was also observed on basket cells, a type of GABAergic interneuron located along the interface between the granule cell layer and the polymorphic layer of the hilus (Figure 1D,G). In the hilus, hilar neurons were intensely labelled for GABA_B1_, showing somatic expression of the subunit (Figure 1E,H). The accumulation of Aβ in APP/PS1 mice was high throughout the molecular layer of the DG, and particularly high in the outer two thirds of this layer compared to age-matched wild type mice (Figure 1I,J). In spite of the large accumulation of Aβ in the DG of APP/PS1 mice, the regional and cellular distributions for GABA_B1_ were very similar between wild and APP/PS1 mice (Figure 1A–H).

### 2.2. Reduction in GABA_B1_ in the Postsynaptic Cell Surface of Granule Cells in APP/PS1 Mice

We recently reported that GABA_B_ receptors are altered in CA1 pyramidal cells in the APP/PS1 mouse model of AD at 12 months of age [17]. To investigate whether other principal cells of the hippocampus are also affected, we used the highly sensitive sodium dodecyl sulfate (SDS)-digested freeze-fracture replica labeling (SDS-FRL)method [34]. To unravel the organisation of GABA_B_ along the surface of granule cells, the subcellular localisation of GABA_B1_ was investigated in sections of the DG obtained from wild type and APP/PS1 mice at 12 months of age. Using our ultrastructural approach, we detected immunoparticles for GABA_B1_ along the plasma membrane of the entire somato-dendritic compartment of granule cells, both in wild type (Figure 2) and APP/PS1 mice (Figure 3), but with notable quantitative differences (Figure 4) as described below.

In wild type mice, immunoparticles for GABA_B1_ were abundantly detected in dendritic spines and dendritic shafts throughout the molecular layer, and also in the somata of granule cells in the granule cell layer (Figure 2A–F). In the six neuronal compartments of granule cells analysed, GABA_B1_ immunoparticles were found forming clusters (an aggregation of more than three gold particles) or scattered (one or two isolated gold particles) (Figure 2A–F). Most immunoparticles were observed in clusters and less frequently scattered in the somata (Figure 2A) and dendritic compartments in the inner one third (Figure 2B–D) and outer two thirds (Figure 2E,F) of the molecular layer. Virtually no labelling was observed on the E-face or on cross-fractures (Figure 2A–F).

In APP/PS1 mice, GABA_B1_ immunoparticles were observed in dendritic spines, dendritic shafts and somata of granule cells (Figure 3A–F). However, the intensity of GABA_B1_ changed significantly in the six neuronal compartments of granule cells analysed. Fewer immunoparticles per cluster and fewer clustered and scattered immunoparticles were detected along the neuronal surface of DG granule cells (Figure 3A–F). Similarly, as with the wild type, no labelling was observed on the E-face or on cross-fractures in APP/PS1 mice (Figure 3A–F).

Next, we performed a quantitative comparison of the GABA_B1_ densities in the six different somato-dendritic compartments of granule cells (Figure 4). In wild type, this analysis showed that GABA_B1_ was not evenly distributed along the neuronal surface in granule cells, but, instead, there was a graded increase in the density of GABA_B1_ immunoparticles from the soma to the dendritic spines (Figure 4). This distribution gradient was also detected in granule cells from APP/PS1 mice (Figure 4). However, our analysis demonstrated that GABA_B1_ density along the neuronal surface was significantly reduced in all compartments analysed in granule cells of the APP/PS1 mice compared to age-matched control mice at 12 months of age (two-way ANOVA test *F*_(1,192)_ = 70.78, *p* < 0.0001 (genotype) and Bonferroni post-hoc test, * *p* < 0.05; ** *p* < 0.01; *** *p* < 0.001) (Figure 4).

### 2.3. Reduction in Presynaptic GABA_B1_ in the Dentate Gyrus of APP/PS1 Mice

In addition to somato-dendritic domains of DG granule cells, immunoparticles for GABA_B1_ were also detected presynaptically in axon terminals, consistent with previous reports [28,29,30,31]. Therefore, we next investigated whether the subcellular distribution of GABA_B1_ at presynaptic sites is altered in the inner one third and outer two thirds of the DG molecular layer of APP/PS1 at 12 months of age using the SDS-FRL technique (Figure 5). In wild type mice, immunoparticles for GABA_B1_ were found forming clusters and scattered outside the clusters in active zones of axon terminals, as well as along the extrasynaptic site of axon terminals, both in the outer two thirds (Figure 5A) and inner one third (Figure 5D) of the DG molecular layer. The density of immunoparticles for GABA_B1_ was higher at the active zones (AZ; inner one third = 288.12 ± 50.29 immunoparticles/µm^2^, *n* = 269 particles; outer two thirds = 379.49 ± 36.27 immunoparticles/µm^2^, *n* = 667 particles) than at extrasynaptic sites (Extra; inner one third = 41.28 ± 11.14 immunoparticles/µm^2^, *n* = 144 particles; outer two thirds = 28.77 ± 6.44 immunoparticles/µm^2^, *n* = 200 particles) (Figure 5C,F). In APP/PS1 mice, immunoparticles for GABA_B1_ showed the same presynaptic distribution in the outer two thirds (Figure 5B) and inner one third (Figure 5E) of the DG molecular layer than that described in wild type mice. However, quantitative comparisons with age-matched wild type mice showed significant differences in the density of GABA_B1_ immunoparticles in the active zone and extrasynaptic sites in the outer two thirds (AZ = 121.30 ± 18.01 immunoparticles/µm^2^, *n* = 76 particles; Extra = 7.86 ± 2.16 immunoparticles/µm^2^, *n* = 33 particles) (Figure 5C), and in the inner one third (AZ = 179.54 ± 37.06 immunoparticles/µm^2^, *n* = 116 particles; Extra = 7.67 ± 1.95 immunoparticles/µm^2^, *n* = 33 particles) (Figure 5F) (for AZ, Two-way ANOVA test *F*_(1,69)_ = 26.48, *p* < 0.0001 (genotype) and Bonferroni post-hoc test, * *p* < 0.05; **** *p* < 0.0001; and for extrasynaptic Two-way ANOVA test *F*_(1,69)_ = 17.09, *p* < 0.0001 (genotype) and Bonferroni post-hoc test, ** *p* < 0.01).. In the two portions of the molecular layer, the reduction in the density of GABA_B1_ was higher in the outer two thirds than in the inner one third (Figure 5C,F), suggesting an input-dependent alteration of GABA_B_ receptors.

### 2.4. GABA_B1_ Is Increased in the Cytoplasm of Granule Cells in APP/PS1 Mice

Finally, to explore the possibility that the reduction in GABA_B1_ along the membrane surface is due to its internalisation and accumulation at intracellular sites of granule cells and axon terminals of APP/PS1 mice at 12 months of age, we used the pre-embedding immunogold technique and quantitative analysis on tissue blocks taken from the inner molecular layer of the DG, where larger differences were detected in the density for GABA_B1_ between wild type and APP/PS1 mice. 

Postsynaptically, immunoparticles for GABA_B1_ were found along the extrasynaptic plasma membrane of dendritic spines and shafts of granule cells, as well as associated at intracellular sites in the same neuronal compartments, both in wild type and APP/PS1 mice (Figure 6). Quantitative analysis showed differences in the plasma membrane versus intracellular sites in the molecular layer (plasma membrane: 42.81% in wild type, *n* = 286 particles, and 18.58% in APP/PS1, *n* = 943 particles; intracellular: 57.19 % in wild type, *n* = 382 particles, and 81.42% in APP/PS1, *n* = 3224 particles) of the DG (Figure 6E). This change in the plasma membrane versus the intracellular membrane labelling was detected both in dendritic spines (plasma membrane: 56% in wild type, *n* = 98 particles, and 35.09% in APP/PS1, *n* = 189 particles; intracellular: 44% in wild type, *n* = 77 particles, and 64.90% in APP/PS1, *n* = 342 particles) and dendritic shafts (plasma membrane: 39.77% in wild type, *n* = 171 particles, and 16.17% in APP/PS1, *n* = 442 particles; intracellular: 60.23% in wild type, *n* = 259 particles, and 83.83% in APP/PS1, *n* = 2278 particles) (Figure 6E). Presynaptically, immunoreactivity for GABA_B1_ was also detected in axon terminals establishing asymmetrical synapses with the dendritic spines of granule cells (Figure 6A–D). The quantitative analysis showed changes in the plasma membrane versus the intracellular membrane among wild type and APP/PS1 in the molecular layer (plasma membrane: 26.98% in wild type, *n* = 17 particles, and 7.65% in APP/PS1, *n* = 26 particles; intracellular: 73.01% in wild type, *n* = 46 particles, and 92.34% in APP/PS1, *n* = 222 particles) (Figure 6E).

## 3. Discussion

The present study provides the first detailed description on the 2D mapping of GABA_B_ receptors along the neuronal surface of DG granule cells, with particular emphasis on their possible alteration in a mouse model of Alzheimer´s disease. Using light microscopy, we showed that GABA_B1_ receptors are widely distributed throughout the different areas of the DG with similar expression intensity and patterns in both wild type and APP/PS1 mice. However, using the SDS-FRL method, our data reveal a decrease in GABA_B_ receptors along the plasma membrane of all subcellular postsynaptic compartments, and a parallel increase at intracellular sites, in DG granule cells at 12 months of age. We additionally reveal a decrease in presynaptic GABA_B_ receptors in axon terminals in the commissural/associational inputs and the entorhinal inputs, both contacting granule cells in a segregated manner. These reductions in postsynaptic and presynaptic GABA_B_ receptors likely contribute to the pathology and memory impairment described in the APP/PS1 model of AD, providing new insights to understand the pathological events taking place in the disease.

### 3.1. Cellular Distribution of GABA_B_ Receptors in the Mouse Model of AD

Dysfunction of the GABAergic system contributes to the modulation of learning and memory mechanism and function [35,36]. Accordingly, the inhibition of GABA_B_ receptors improves learning and memory formation [37,38] and altered GABA_B_ receptor function has been involved in AD pathogenesis [39]. Previous autoradiographic, *in situ* hybridisation and immunohistochemical reports showed that DG contains high levels of GABA_B_ receptors [20,26,28,29]. At the cellular level, granule cells express transcripts for both GABA_B1a_ and GABA_B1b_ subunits [27], although they are targeted to different synaptic sites to perform distinct functions [30]. Thus, mossy fibre terminals are immunopositive for the GABA_B1a_ subunit, whereas dendrites of granule cells are immunopositive for GABA_B1b_ [30,40]. In the present study, we used a pan-GABA_B1_ antibody and, therefore, it was not possible to differentiate between the expression and distribution of both GABA_B1a_ and GABA_B1b_ subunits in granule cells. However, immunoreactivity for the GABA_B_ receptor subunit was predominantly enriched in the molecular layer of the DG, suggesting a mainly dendritic localisation.

Growing evidence shows that dysfunction of GABA_B_-mediated synaptic transmission underlies a number of disorders in the brain [39,41,42,43]. Given the role of GABA_B_ receptors in learning and memory [37,42], their involvement in AD has also been suggested. Supporting this idea, recent studies reported not only that GABA_B_ receptors selectively co-purify with APP [15], but also that GABA_B_-APP macromolecular complexes link presynaptic GABA_B_ receptor trafficking to Aβ formation [16]. In addition, autoradiography studies of GABA_B_ receptor expression in the human hippocampus of AD patients demonstrated fewer GABA_B_ receptors in the molecular layer of the dentate gyrus [44]. Here, we described that the regional and cellular distribution of GABA_B_ receptors in the DG is virtually identical in APP/PS1 and control mice. Similarly, we have recently described that the expression of GABA_B_ receptors does not change in the CA1 and CA3 regions of the hippocampus in the same mouse model of AD, although expression is significantly reduced in the human hippocampus of AD patients [17].

These discrepancies could be likely due to differences in brain organisation between humans and mice [45], to the fact that the APP/PS1 model does not develop neurofibrillary tangles, a typical pathologic alteration observed in AD patients [46], or simply because the age of 12 months old used here, as well as by other laboratories, is still too short to detect gross changes. However, this mouse model shows reduced number of synapses per volume and the synaptic morphology in plaque-free regions of the DG outer molecular layer [10], and thus it is useful to detect possible alterations of synapse-associated receptors and/or ion channels [17].

### 3.2. Altered Somato-Dendritic Localisation of GABA_B_ Receptors in APP/PS1 Granule Cells

Using the SDS-FRL technique, we detected the majority of postsynaptic GABA_B1_ immunoparticles along the extrasynaptic membrane of dendritic spines and shafts of granule cells, consistent with our immunoreactions at the light microscopic level. GABA_B_ receptors localised to postsynaptic sites activate G protein-gated inwardly rectifying K+ (GIRK) channels and are responsible for the slow inhibitory postsynaptic potentials IPSP [47,48], and GABA_B_ receptor-mediated slow inhibitory responses have been found in granule cells [31,49,50,51,52]. The prevalent localisation of GABA_B1_ in spines and dendrites we revealed here is in agreement with those electrophysiological observations showing the postsynaptic role of GABA_B_ receptors in granule cells. Interestingly, GIRK channels show a very similar somato-dendritic distribution than GABA_B_ receptors [31], suggesting that those GABA_B_-GIRK signalling cascade controls excitability in granule cell membranes.

The laminar arrangement of the cell bodies of granule cells in the DG allowed us to investigate changes in the density of GABA_B_ receptors in dendritic spines as a function of distance from the soma in wild type and APP/PS1 mice. The molecular layer is occupied mainly by the dendritic arbour of granule cells, as well as axons and terminal axonal arbours from the entorhinal cortex and other sources. Granule cells receive a major input from cells located in layer II of the entorhinal cortex, via the so-called perforant pathway. The entorhinal terminals are strictly confined to the outer two-thirds of the molecular layer where they form asymmetric synapses with spines of granule cells [53]. Inputs forming the commissural/associational pathway are limited to the inner one third of the molecular layer and largely contain axon terminals from the ipsilateral and contralateral hippocampus [53].

Given this anatomical organisation, quantitative comparison of immunogold densities revealed the distance- and subcellular compartment-specific distribution pattern of GABA_B_ receptors on the surface of granule cells. Our high-resolution immunolocalisation experiments showed an increase in immunolabelling for GABA_B1_ from proximal to distal dendrites. Consistent with our morphological data, electrophysiological studies have shown that GABA_B_ receptor-mediated currents are significantly larger in dendrites than in the somata of granule cells, and were markedly higher for the distal primary dendrites in the outer molecular layer [31]. Interestingly, GIRK channels show a very similar somato-dendritic distribution than GABA_B_ receptors [31], suggesting that the GABA_B_-GIRK signalling cascade controls excitability in granule cell membranes.

In APP/PS1 mice, our analysis, performed at 12 months of age, also revealed distance-dependent and subcellular compartment-specific localisation of GABA_B_ receptors in granule cells. However, the main finding of this study is the significant reduction in the membrane surface distribution of GABA_B_ receptors in all subcellular compartments of granule cells. These findings are consistent with the evidence that the density of binding sites for GABA_B_ receptors is decreased in molecular layer of the DG in human hippocampus with AD [44]. In addition, parallel to the decrease in the density of GABA_B1_ along the membrane surface, the application of different immunoelectron microscopy and quantitative analyses showed that there is an accumulation of immunoparticles at cytoplasmic sites in the dendrites and spines of granule cells. This pool of internalised GABA_B_ receptors does not seem to be redirected to lysosomal degradation in AD, not yet, at least, at 12 months of age. A similar internalisation has been recently described in CA1 pyramidal cells in the same mouse model [17], suggesting that different principal cells in the hippocampus use similar mechanisms to remove GABA_B_ receptors from the plasma membrane.

### 3.3. Altered Presynaptic GABA_B_ Receptors in the Molecular Layer of APP/PS1 Mice

In addition to the main presence of GABA_B1_ in somato-dendritic compartments of granule cells, we also found presynaptic labelling for this receptor subunit in the axon terminals establishing excitatory synapses in the molecular layer of the DG. Presynaptic labelling for GABA_B_ receptors has been described in other regions of the hippocampus, including the CA1 and CA3 subfields [17,28,29,30,54]. Presynaptic activation of GABA_B_ receptors mediate slow and prolonged synaptic inhibition by inhibiting voltage-activated calcium (Cav) channels to result primarily in reduced vesicular release of neurotransmitter [13]. Using the SDS-FRL technique, we observed immunoparticles for GABA_B1_ along the extrasynaptic membrane and the active zone of axon terminals located both in the outer two thirds and inner one third of the molecular layer, in agreement with previous studies using pre-embedding immunogold techniques [28,29]. However, in the mouse model of AD we not only found a reduction in the density of GABA_B_ receptors in the inner and outer molecular layer, but also that the decrease was strongest in the termination area of the perforant pathway. Overall, these results are in good agreement with studies showing that the increase in Aβ production is linked to dysfunctional axonal trafficking and to the reduction in the expression of GABA_B_ receptor in the APP knockout mice [16].

The functional consequences of this input-dependent reduction in GABA_B_ receptors is not yet known, but our data agrees with previous reports showing that AD pathology begins in the entorhinal cortex and spreads through the perforant pathway to the hippocampus [55]. Since the DG is a hippocampal region with a large accumulation of Aβ in AD [56], the larger decrease in the density of presynaptic GABA_B_ receptors in the outer two thirds of the molecular layer may reflect changes due to entorhinal cortical pathology. However, since the perforant pathway lacks strong presynaptic GABA_B_ receptor-mediated inhibition [57], the main effect produced by the reduction in GABA_B_ receptor expression in the APP/PS1 mice mostly arises from postsynaptic receptors. Altogether, given the cognitive deficits observed in APP/PS1 mice [58], the pre- and post-synaptic reduction in GABA_B_ receptors in granule cells, together with the alterations affecting other principal cells of the hippocampus [17], may represent the molecular and anatomical substrate of the altered cognitive function observed in this mouse model of AD. Supporting this idea, it has been shown that CGP55845, an antagonist of GABA_B_ receptors, restored the memory of mice and augmented LTP [59]. More recently, the use of a different GABA_B_ receptor antagonist, CGP35348, showed beneficial effects in improving cognition, learning, and memory in memory tests in acute Aβ toxicity-induced rats [60]. These data indicate that potential therapeutics may target GABA_B_ receptors in the hippocampus.

## 4. Material and Methods

### 4.1. Animals

The mouse line used for this study (APP/PS1; hemizygote animals) expressed the Mo/Hu APP695swe construct in conjunction with the exon-9-deleted variant of human presenilin 1 (PS1-dE9) [61]. Control mice were age-matched littermates without the transgene (wild type). Mice of all genotypes were aged to 12 months before use in a battery of morphological experiments. For each genotype, four mice were used for SDS-FRL techniques, three mice were used for immunohistochemistry at the light microscopic level and three mice were used for pre-embedding immunoelectron microscopic analyses. All mice were obtained from the Animal House Facility of the University of Castilla–La Mancha (Albacete, Spain). Animals were housed in cages of two or more mice, maintained on a 12 h light/12 h dark cycle at 24°C and received food and water ad libitum. Both the care and handling of animals prior to and during experimental procedures were carried out in accordance with Spanish (RD 1201/2015) and European Union regulations (86/609/EC), and the protocols were approved (22 March 2019) by the local Animal Care and Use Committee.

For immunohistochemistry at the light and electron microscopic levels, animals were deeply anaesthetised by intraperitoneal injection of ketamine-xylazine 1:1 (0.1 mL/kg) or sodium pentobarbital (50 mg/kg, i.p.) and transcardially perfused with ice-cold fixative containing 1) 2% paraformaldehyde in 0.1 M phosphate buffer (PB, pH 7.4) for 12 min (for SDS-FRL technique), or 2) 4% paraformaldehyde, 0.05% glutaraldehyde, and 15% (*v*/*v*) saturated picric acid made up in 0.1 M phosphate buffer (PB), pH 7.4 (for light microscopy and pre-embedding immunogold techniques). After perfusion, these brains were removed and coronal sections were cut on a Vibratome (Leica VT1000).

### 4.2. Antibodies and Chemicals

In the present study, we used the following primary antibodies: 1) An affinity-purified rabbit polyclonal antibody representing the amino acid sequence of GABA_B1_ (ref#B17, residues 901–960 of rat GABA_B1_). This antibody has been characterised and the specificity proven elsewhere [30,62,63]; and 2) An affinity-purified rabbit polyclonal antibody anti-β amyloid (ref #2454, detecting both human Aβ-40 and Aβ-42 peptides; Cell Signalling Technology, Leiden, The Netherlands). In addition, we used the following secondary antibodies: alkaline phosphatase (AP)-goat anti-rabbit IgG (H+L) diluted 1:5000 (Invitrogen, Paisley, UK), goat anti-rabbit IgG coupled to 1.4 nm gold diluted 1:100 (Nanoprobes Inc., Stony Brook, NY, USA) and anti-rabbit IgG conjugated to 10 nm gold particles diluted 1: 100 (British Biocell International, Cardiff, UK).

### 4.3. Immunohistochemistry for Light Microscopy

The immunoperoxidase method was chosen to perform immunohistochemical reactions at the light microscopic level, as we described in detail previously [64]. Hippocampal sections were incubated for one hour in 10% normal goat serum (NGS) diluted in 50 mM Tris buffer (pH 7.4) containing 0.9% NaCl (TBS) and 0.2% Triton X-100. After several washes in TBS, hippocampal sections were incubated in the primary antibody (anti-GABA_B1_ or anti-β amyloid, both at a concentration of 1–2 µg/mL and diluted in TBS containing 1% NGS). Next, the sections were incubated in biotinylated goat anti-rabbit IgG (Vector Laboratories, Burlingame, CA, USA) diluted 1:200 in TBS containing 1% NGS, and then transferred into avidin–biotin–peroxidase complex (ABC kit, Vector Laboratories). Next, we revealed peroxidase enzyme activity using the chromogen 3,3´-diaminobenzidine tetrahydrochloride (DAB; 0.05% in TB, pH 7.4) and the substrate H_2_O_2_ (0.01% in distilled water). Hippocampal sections were then air-dried for few hours, mounted in DPX (Merk, Darmstadt, Germany), coverslipped and observed in a Leica photomicroscope (DM2000) equipped with differential interference contrast optics and a digital camera (DFC500).

### 4.4. Immunohistochemistry for Electron Microscopy

The pre-embedding immunogold and the SDS-digested freeze-fracture replica labelling (SDS-FRL) techniques were chosen to perform immunohistochemical reactions at the electron microscopic level, as we described in detail previously [34,64,65].

#### 4.4.1. Pre-Embedding Immunogold Method

Hippocampal sections obtained from both WT and APP/PS1 mice at 12 months of age were incubated in 10% NGS diluted in TBS.

After several washes in TBS, hippocampal sections were incubated in the primary antibody (anti-GABA_B1_ at a concentration of 3–5 µg/mL and diluted in TBS containing 1% NGS). After several washes in TBS, the sections were incubated in goat anti-rabbit IgG coupled to 1.4 nm gold (Nanoprobes Inc., Stony Brook, NY, USA). Next, hippocampal sections were postfixed in 1% glutaraldehyde, washed in double-distilled water and silver enhancement of the gold particles (HQ Silver kit, Nanoprobes Inc.). Following treatment with osmium tetraoxide (1% in 0.1 M phosphate buffer), block-staining with uranyl acetate and dehydration in graded series of ethanol, the sections were then flat-embedded on glass slides in Durcupan (Fluka) resin. After polymerization of the resin, regions of interest were cut at 70–90 nm on an ultramicrotome (Leica EM UC7, Leica, Wetzlar, Germany) and collected on pioloform-coated copper grids. Finally, ultrathin sections were stained on drops of 1% aqueous uranyl acetate and Reynolds’s lead citrate. For ultrastructural analyses we used a JEOL-1010 electron microscope.

#### 4.4.2. SDS-FRL Technique

Mice brains were perfused using 2% paraformaldehyde in 0.1 M phosphate buffer (PB) for 12 min. Following several washes in 0.1 M PB, the hippocampus was dissected and cut sagittal (130 µm-thick sections) using a Microslicer (Dosaka, Kyoto, Japan). Hippocampal sections containing the DG were further trimmed down, immersed in graded glycerol of 10%–30% in 0.1 M PB at 4 °C overnight, and frozen using a high-pressure freezing machine (HPM010, BAL-TEC, Balzers, Liechtenstein). Next, DG slices were fractured into two parts at −120 °C and then replicated by depositions of carbon (5 nm-thick), platinum (60° unidirectional from horizontal level, 2 nm-thick), and carbon (15–20 nm-thick) using a freeze-fracture replica machine (BAF060, BAL-TEC, Balzers). The resulting replicas were then transferred to 2.5% SDS and 20% sucrose in 15 mM Tris buffer (pH 8.3) for 18 h at 80 °C with shaking, washed three times in 50 mM TBS (pH 7.4) containing 0.05% bovine serum albumin (BSA), and blocked with 5% BSA for 1 h at room temperature. After several washes in TBS containing 0.05% BSA, replicas were incubated in a polyclonal rabbit antibody for GABA_B1_ (5 μg/mL) at 15 °C overnight. Next, replicas were blocked in 5% BSA/TBS and incubated in secondary antibodies conjugated with 10 nm gold particles overnight at room temperature. Finally, the replicas were rinsed with 0.05% BSA in TBS, washed with distilled water, and picked up onto grids coated with pioloform (Agar Scientific, Stansted, Essex, UK). For ultrastructural analyses, we used a JEOL-1010 electron microscope.

### 4.5. Quantification and Analysis of SDS-FRL Data

The labelled replicas were examined using a transmission electron microscope (JEOL-1010) and photographed at magnifications of 60,000, 80,000, and 100,000×. All antibodies used in this study were visualised by immunoparticles on the protoplasmic face (P-face), consistent with the intracellular location of their epitopes. Non-specific background labelling was measured on E-face surfaces in wild type mice. Digitised images were then modified for brightness and contrast using Adobe PhotoShop CS5 (Mountain View, CA, USA) to optimise them for quantitative analysis of immunolabelling. The quantitative analyses were done using the software Gold Particle Detection and Quantification (GPDQ), developed recently to perform automated and semi-automated detection of gold particles present in a given compartment of neurons [34].

#### Density Gradient of GABA_B1_ along the Neuronal Surface

The procedure was similar to that used previously [34]. Briefly, immunogold labelling for GABA_B1_ was achieved from replicas containing all three layers of the DG, so that the laminar distribution could be compared under identical conditions for each animal and experimental group. Quantitative analysis of immunogold labelling for GABA_B1_ was performed on five different dendritic compartments of granule cells in the molecular layer of the DG and in the somata of granule cells in the granule cell layer. The dendritic compartments analysed were the main dendritic shaft (primary dendrites), spiny branchlets (secondary dendrites) and dendritic spines. Secondary dendrites were identified based on their small diameter and the presence of at least one emerging spine from the dendritic shaft. Dendritic spines were considered as such if: (i) they emerged from a dendritic shaft, or (ii) they opposed an axon terminal. Axon terminals were identified based on: (i) the presence of an active zone (AZ) facing a postsynaptic density (PSD), recognised by an accumulation of intramembrane particles (IMPs), on the opposing exoplasmic-face (E-face) of a spine or dendrite with spines; or (ii) the presence of synaptic vesicles on their cross-fractured portions. Non-specific background labelling was measured on E-face structures surrounding the measured P-faces. Images of the identified compartments were selected randomly over the entire dendritic tree of granule cells and then captured with an ORIUS SC1000 CCD camera (Gatan, Munich, Germany). The area of the selected profiles and the number of immunoparticles were measured using our GPDQ software [34]. Immunoparticle densities were presented as mean ± SEM between animals. Statistical comparisons were performed with GraphPad software (San Diego, CA, USA).

### 4.6. Quantification and Analysis of Pre-Embedding Immunogold Data

To establish the relative abundance of GABA_B1_ immunoreactivity in different compartments of granule cells between the two genotypes, pre-embedding immunogold immunohistochemistry was carried out [64]. Briefly, for each of three adult animals, three samples of tissue were obtained for the preparation of embedding blocks, thus using in total nine blocks. To minimise false negatives, electron microscopic serial ultrathin sections were cut close to the surface of each block, as immunoreactivity decreased with depth. We estimated the quality of immunolabelling by always selecting areas with optimal gold labelling at approximately the same distance from the cutting surface. Randomly selected areas were then photographed from the selected ultrathin sections and used with final magnification between 30,000 and 50,000 ×. Quantification of immunogold labelling was carried out in reference areas of the DG totalling approximately 2500 µm^2^. We counted immunoparticles identified in each reference area and present in different subcellular compartments: dendritic spines, dendritic shafts, and axon terminals. The data were expressed as a percentage of immunoparticles in each subcellular compartment, both in the plasma membrane and at intracellular sites.

Finally, it is worth mentioning that neuronal loss is only observed adjacent to plaques in the APP/PS1 mouse model [10]. It should be noted that both in mouse models of AD and in AD patients, neurons in contact with Aβ plaques suffer alterations in the morphology and number of dendritic spine, with destroyed tissue in dystrophic neurites [66,67,68,69], as well as in synaptic transmission [55,67,68,70,71]. To avoid this situation, our quantitative analysis was performed in Aβ plaque-free regions of the DG. Thus, the density values expressed as immunoparticles/µm^2^ in the APP/PS1 mice represent a genuine reduction in GABA_B_ receptors in different compartments of granule cells, regardless of any possible neuronal and/or synaptic loss.

### 4.7. Controls

To test method specificity in the procedures for electron microscopy, the primary antibody was either omitted or replaced with 5% (*v*/*v*) normal serum of the species of the primary antibody, resulting in total loss of the signal. For the pre-embedding technique, labelling patterns were also compared with those obtained with Calbindin (polyclonal rabbit anti-Calbindin D-9k CB9; Swant, Marly, Switzerland); only the antibodies against GABA_B1_ consistently labelled the plasma membrane.

### 4.8. Data analysis

Statistical analyses for morphological data were performed using SigmaStat Pro (Jandel Scientific) and data were presented as mean ± SEM unless indicated otherwise. Statistical significance was defined as *p* < 0.05. The statistical evaluation of the immunogold densities was performed using the two-way ANOVA test, and further compared with the Bonferroni post-hoc test.

## Figures and Tables

**Figure 1 ijms-21-02459-f001:**
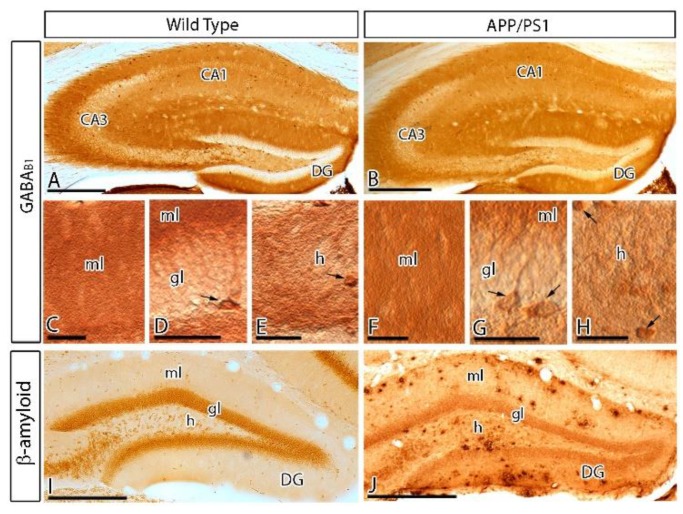
Regional and cellular distribution of the metabotropic γ-aminobutyric acid GABA_B1_ subunit (GABA_B1_) in wild type and APP/PS1 mice. (**A**–**H**) Immunoreactivity for GABA_B1_ in the dentate gyrus (DG) of wild type and APP/PS1 mice at 12 months of age using an immunohistochemical method at the light microscopic level. Immunoreactivity for GABA_B1_ was very similar both in the wild type and the APP/PS1 mice. Labelling for GABA_B1_ showed the highest intensity in molecular layer (ml) of the DG and weaker in the granule cell layer (gl). Immunoreactivity for GABA_B1_ was also detected in pyramidal basket cells (arrows) located in the granule cell layer and hilar neurons (arrows located in the hilus (h)), with similar distribution pattern and labelling intensity in wild type and APP/PS1 mice. (**I**,**J**) Immunoreactivity for β-amyloid in wild type and APP/PS1 mice, showing high accumulation of Aβ throughout all layers of the DG, and particularly high expression in the outer molecular layer. Abbreviations: CA1 region of the hippocampus (CA1); CA3 region of the hippocampus (CA3); dentate gyrus (DG); molecular layer (ml); granule cell layer (gc); hilus (h). Scale bars: **A**,**B**,**I**,**J** 200 µm; **C**–**H**, 30 µm.

**Figure 2 ijms-21-02459-f002:**
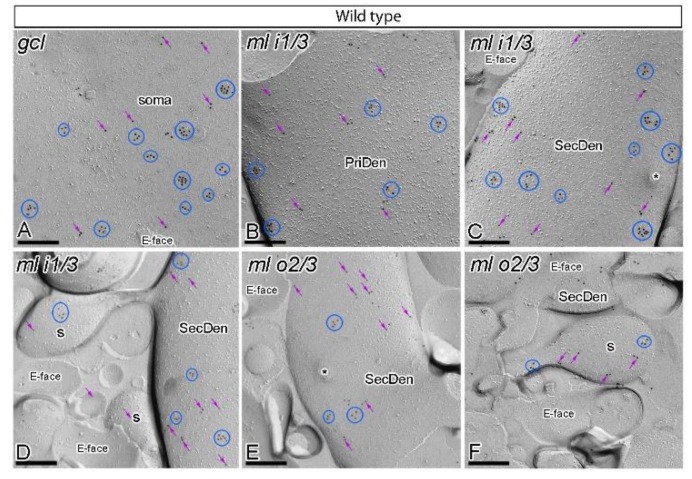
Subcellular localisation of GABA_B1_ in somato-dendritic domains of granule cells in wild type mice. (**A**–**F**) Electron micrographs obtained from different parts of the DG showing immunoparticles for GABA_B1_ along the surface of granule cells, as detected using the SDS-FRL technique in wild type mice at 12 months of age. Immunoparticles for GABA_B1_ were detected forming clusters (blue ellipses/circles) or scattered (purple arrows) associated with the P-face in the soma, primary dendrites (PriDen), secondary dendrites (SecDen) and dendritic spines (s) of both the outer two thirds (ml o2/3) and inner one third (ml i1/3) of the molecular layer. Fractured spine necks are indicated with asterisks (*). The E-face is free of any immunolabelling. Scale bars: **A**–**F**, 0.2 μm.

**Figure 3 ijms-21-02459-f003:**
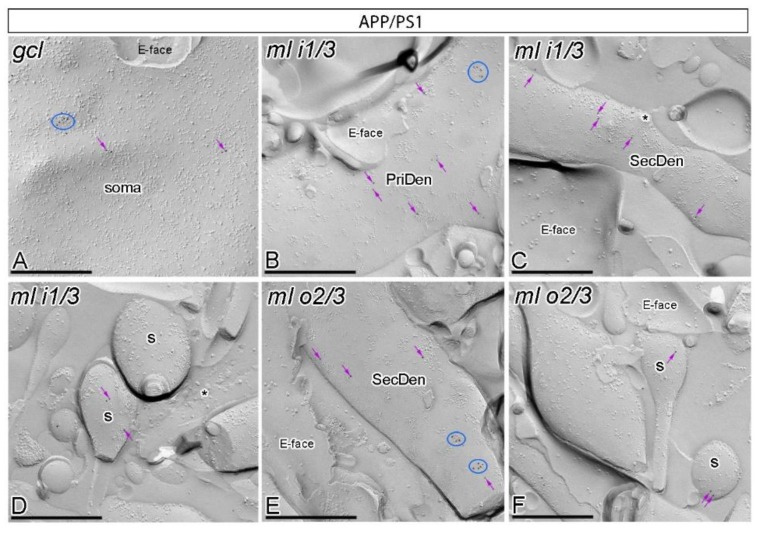
Subcellular localisation of GABA_B1_ in somato-dendritic domains of granule cells in APP/PS1 mice. (**A**–**F**) Electron micrographs obtained from different parts of the DG showing immunoparticles for GABA_B1_ along the surface of granule cells, as detected using the SDS-FRL technique in APP/PS1 mice at 12 months of age. Immunoparticles for GABA_B1_ were detected at lower frequency forming clusters (blue ellipses/circles) or scattered (purple arrows) associated with the P-face in the soma, primary dendrites (PriDen), secondary dendrites (SecDen) and dendritic spines (s) of both the outer two thirds (ml o2/3) and inner one third (ml i1/3) of the molecular layer. Fractured spine necks are indicated with asterisks (*). The E-face is free of any immunolabelling. Scale bars: **A**–**F**, 0.2 μm.

**Figure 4 ijms-21-02459-f004:**
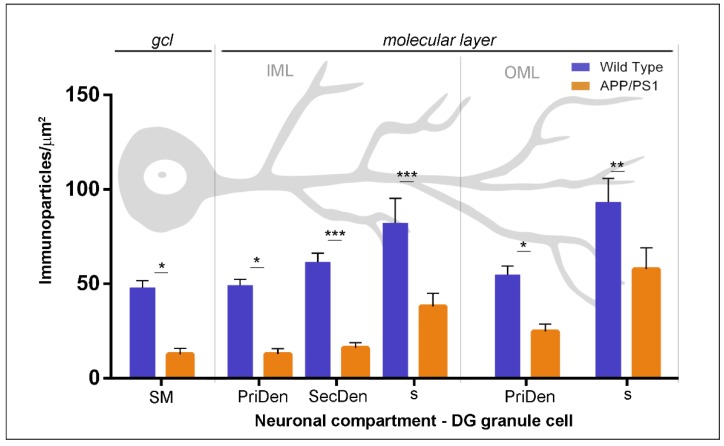
Density gradient of GABA_B1_ immunoparticles along the surface of granule cells in wild type and APP/PS1 mice at 12 months. Quantitative analysis of GABA_B1_ immunogold labelling in six neuronal compartments of DG granule cells. The density gradient of immunoparticles for GABA_B1_ along the neuronal surface was significantly reduced in all subfields and compartments of granule cells analysed in the APP/PS1 mice compared to age-matched wild type (Two-way ANOVA test *F*_(1,192)_ = 70.78, *p* < 0.0001 (genotype) and Bonferroni post-hoc test, * *p* < 0.05; ** *p* < 0.01; *** *p* < 0.001). Error bars indicate SEM. Abbreviations: PriDen, primary dendrite; SecDen, secondary dendrite; s, spine; SM, soma.

**Figure 5 ijms-21-02459-f005:**
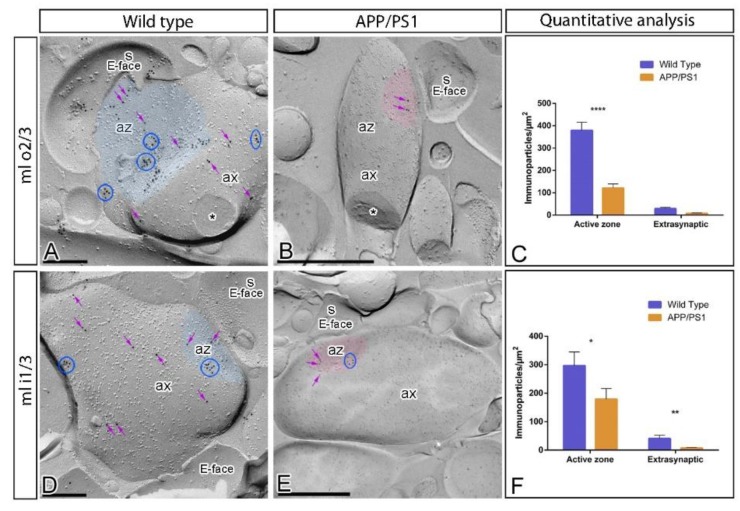
Presynaptic distribution of GABA_B1_ in the dentate gyrus in wild type and APP/PS1 mice. Electron micrographs showing immunoparticles for GABA_B1_ in presynaptic compartments in the outer two thirds (ml o2/3) and inner one third (ml i1/3) of the molecular layer of the DG at 12 months of age, as detected using the SDS-FRL technique. (**A**,**D**) In wild type, immunoparticles for GABA_B1_ were found forming clusters (blue ellipses/circles) and also detected scattered (purple arrows) outside the clusters within the active zone (az, blue transparent overlay) and along the extrasynaptic site of axon terminals (ax). (**B**,**E**) In APP/PS1, fewer immunoparticles for GABA_B1_ forming clusters (blue ellipses/circles) or scattered (purple arrows), were detected within the active zone (az, red transparent overlay) and along the extrasynaptic plasma membrane of axon terminals (ax). Cross-fractures are indicated with asterisks (*). (**C**,**F**) Histograms illustrating densities of immunoparticles for GABA_B1_ in presynaptic compartments in the outer two thirds (ml o2/3) and inner one third (ml i1/3) of the molecular layer in wild type and APP/PS1 mice. Significant differences were detected in densities of GABA_B1_ immunoparticles in the outer two thirds of the molecular layer (Wild type: Active zone (AZ) = 379.49 ± 36.27 immunoparticles/µm^2^ and Extra = 28.77 ± 6.44 immunoparticles/µm^2^; APP/PS1: AZ = 121.31 ± 18.01 immunoparticles/µm^2^; Extra = 7.87 ± 2.16 immunoparticles/µm^2^), as well as in the inner one third of the molecular layer (Wild type: AZ = 288.12 ± 50.29 immunoparticles/µm^2^ and Extra = 41.28 ± 11.14 immunoparticles/µm^2^; APP/PS1: AZ = 179.54 ± 37.06 immunoparticles/µm^2^; Extra = 7.67 ± 1.94 immunoparticles/µm^2^) (for AZ, Two-way ANOVA test *F*_(1,69)_ = 26.48, *p* < 0.0001 (genotype) and Bonferroni post-hoc test, * *p* < 0.05; **** *p* < 0.0001; and for extrasynaptic Two-way ANOVA test *F*_(1,69)_ = 17.09, *p* < 0.0001 (genotype) and Bonferroni post-hoc test, ** *p* < 0.01). Error bars indicate SEM. Scale bars: **A**,**D**, 0.2 μm; **B**,**E**, 0.5 μm.

**Figure 6 ijms-21-02459-f006:**
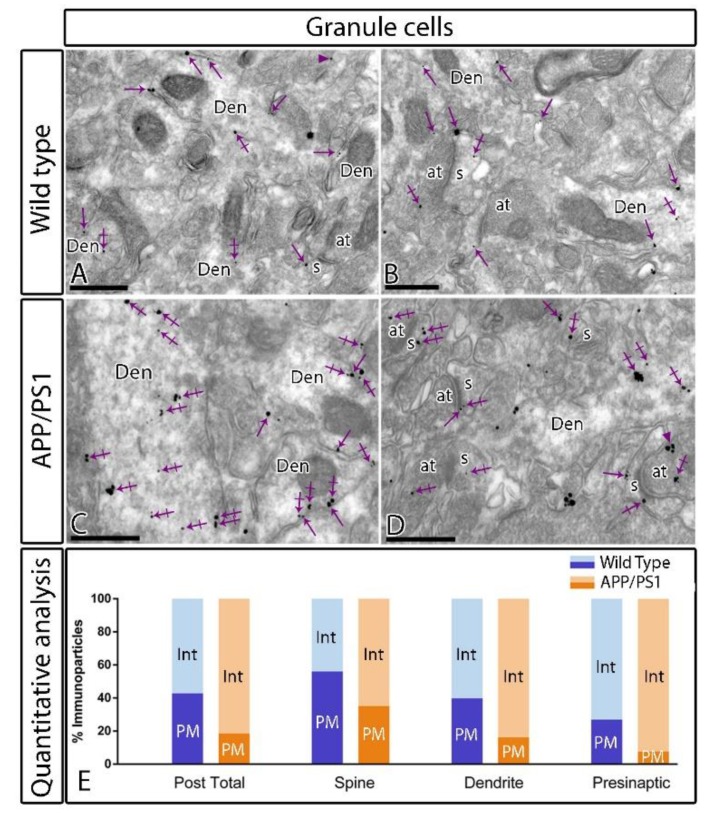
Intracellular distribution of GABA_B1_ is increased in the DG of APP/PS1 mice. Electron micrographs showing immunoparticles for GABA_B1_ in granule cells of the inner molecular layer of the DG at 12 months of age in wild type and APP/PS1 mice, as detected using a pre-embedding immunogold technique. (**A**–**D**) Both in wild type and APP/PS1 mice, GABA_B1_ immunoparticles were found along the extrasynaptic plasma membrane (arrows) and intracellular sites (crossed arrows) of dendritic shafts (Den) and spines (s) of granule cells contacted by axon terminals (at), and less frequently at presynaptic sites (arrowheads). (**E**) Quantitative analysis showed that immunoparticles for GABA_B1_ were less frequently observed along the extrasynaptic plasma membrane dendrites and spines of granule cells in APP/PS1 mice, as well at presynaptic sites, but, instead, they were more frequently detected at intracellular sites. Scale bars: **A**–**D**, 0.4 μm.

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
