# Peer review of "Density of GABAB Receptors Is Reduced in Granule Cells of the Hippocampus in a Mouse Model of Alzheimer’s Disease"

_ijms, 2020, doi:10.3390/ijms21072459_

Round 1
Reviewer 1 Report
The authors investigated the distribution of GABA-B1 subunit in the dentate gyrus using a in vivo model of Alzheimer’s Disease. They observed a presynaptic and postsynaptic reduction of GABA-B1 subunit in the granule cells of the dentate gyrus in AD model mice compared to controls. These data are interesting and well correlated with previous reports. The paper is well written and suitable for the publication.
Author Response
We are glad to know that this Referee thinks that our results are interesting and the manuscript is suitable for publication in the International Journal of Molecular Sciences.
Reviewer 2 Report
The paper provides novel, interesting and important data on the expression of GABAB receptor in animal model of AD.
The results are clearly described, but the immunohistochemical pictures could be improved. I would suggest to enlarge pictures included in the composition of Fig.1, as at present it is difficult to see the details.
On Figs 2, 3, 5 and 6 I would suggest to use font color different than black as the contrast between the background and the font is weak. The same is with arrows. Please, try to use red, yellow or something like this to make it clearly seen for the readers.
When you use two-way ANOVA, F values should be provided.
The paper is strongly histological, and the methodolody description and detailed staining are perfectly described. However, I would suggest to shorten the discussion on this and to make it easier for the reader, I suggest to compose a table with the summarisation of the staining.
Also, there is limited data supporting the thesis proposed by the authors. Please, include more data from human post mortem studies, if they are available, and also results from animal studies on the activity of GABAB ligands in animal models of AD.
Author Response
We are grateful for the valuable suggestions of the Referee that we have incorporated into the manuscript. The changed words have been highlighted in bold in the manuscript.
Q 1 - Specific comment: “The results are clearly described, but the immunohisto-chemical pictures could be improved. I would suggest to enlarge pictures included in the composition of Fig.1, as at present it is difficult to see the details”.
Authors’ response: Following the suggestion of the reviewer, we have enlarged all panels included in Figure 1, and particular we have significantly enlarged the three panels for each genotype including details of the dentate gyrus.
Q 2 - Specific comment: “On Figs 2, 3, 5 and 6 I would suggest to use font color different than black as the contrast between the background and the font is weak. The same is with arrows. Please, try to use red, yellow or something like this to make it clearly seen for the readers”.
Authors’ response: Following the suggestion of the reviewer, we have modified fronts in the different Figures. We tried changing colours of font, but could not find out a good colour that was really visible; red or yellow colour did not provide appropriate visibility. Instead, we have surrounded the font with a white line, finding out that fonts were now more visible and differentiated from background of panels. Finally, and also following the suggestion of the reviewer, we have changed the black colour of arrows in Figure 6 to a magenta colour.
Q 3 - Specific comment: “When you use two-way ANOVA, F values should be provided”.
Authors’ response: Following the suggestion of the reviewer, we have provided F values, which have been incorporated in figure legends of Figure 4 and Figure 5.
Q 4 - Specific comment: “The paper is strongly histological, and the methodology description and detailed staining are perfectly described. However, I would suggest to shorten the discussion on this and to make it easier for the reader, I suggest to compose a table with the summarisation of the staining”.
Authors’ response: Following the suggestion of the reviewer, we have shortened the discussion section by removing the whole heading about “Methodological considerations”, corresponding to the removal of 463 words. Part of the last paragraph of that heading was included in pages 10-11 of Material and methods.
Q 5 - Specific comment: “Also, there is limited data supporting the thesis proposed by the authors. Please, include more data from human post mortem studies, if they are available, and also results from animal studies on the activity of GABAB ligands in animal models of AD”.
Authors’ response: Data on the expression of neurotransmitter receptors or ion channels in human post mortem samples is very limited, and this is particularly true for GABAB receptors. Nevertheless, following the suggestion of the reviewer, we have added to the discussion a few more sentences (highlighted in bold) in different paragraphs to meet the requirement of the reviewer.